# Novel Beta Lactam Antibiotics for the Treatment of Multidrug-Resistant Gram-Negative Infections in Children: A Narrative Review

**DOI:** 10.3390/microorganisms11071798

**Published:** 2023-07-13

**Authors:** Francesco Venuti, Lorenza Romani, Maia De Luca, Costanza Tripiciano, Paolo Palma, Maria Chiriaco, Andrea Finocchi, Laura Lancella

**Affiliations:** 1Unit of Infectious Diseases, Department of Medical Sciences, University of Torino, Amedeo di Savoia Hospital, 10149 Torino, Italy; francescovenuti1993@gmail.com; 2Infectious Disease Unit, Bambino Gesù Children’s Hospital, IRCCS, 00165 Rome, Italy; lorenza.romani@opbg.net (L.R.); maia.deluca@opbg.net (M.D.L.); costanza.tripiciano@opbg.net (C.T.); laura.lancella@opbg.net (L.L.); 3Unit of Clinical Immunology and Vaccinology, Bambino Gesù Children’s Hospital, IRCCS, 00165 Rome, Italy; paolo.palma@opbg.net; 4Department of Systems Medicine, University of Rome Tor Vergata, 00133 Rome, Italy; andrea.finocchi@opbg.net; 5Research Unit of Primary Immunodeficiencies, Bambino Gesù Children’s Hospital, IRCCS, 00165 Rome, Italy

**Keywords:** children, pediatric, multidrug resistant, Gram-negative bacteria, infections, ceftolozane/tazobactam, ceftazidime/avibactam, meropenem/vaborbactam, imipenem/relebactam, cefiderocol

## Abstract

Infections due to carbapenem-resistant Enterobacterales (CRE) are increasingly prevalent in children and are associated with poor clinical outcomes, especially in critically ill patients. Novel beta lactam antibiotics, including ceftolozane-tazobactam, ceftazidime-avibactam, meropenem-vaborbactam, imipenem-cilastatin-relebactam, and cefiderocol, have been released in recent years to face the emerging challenge of multidrug-resistant (MDR) Gram-negative bacteria. Nonetheless, several novel agents lack pediatric indications approved by the Food and Drug Administration (FDA) and the European Medicine Agency (EMA), leading to uncertain pediatric-specific treatment strategies and uncertain dosing regimens in the pediatric population. In this narrative review we have summarized the available clinical and pharmacological data, current limitations and future prospects of novel beta lactam antibiotics in the pediatric population.

## 1. Introduction

The proliferation of multidrug-resistant (MDR) Gram-negative bacteria, such as extended-spectrum β-lactamase (ESBL)–producing *Escherichia coli*, *Klebsiella pneumoniae*, and *Pseudomonas aeruginosa*, is a major global threat for the health of our population, being associated with less favorable outcomes [1,2,3]. In this scenario, selecting the appropriate antimicrobial therapy has become increasingly challenging for clinicians, and the need for new therapeutic options has become urgent [4].

The pediatric population is subject to high antibiotic exposures with global estimates indicating that between 37 and 60% of hospitalized pediatric patients receive at least one antibiotic highlighting the worldwide relevance of the problem [5,6]. As the use of antimicrobials is a known driver for resistance [7], an increasing incidence of MDR Gram-negative infections has been observed in infants and children [8,9,10,11,12]. To overcome resistance mechanisms, novel beta lactam agents including ceftolozane/tazobactam (C/T), ceftazidime/avibactam, meropenem/vaborbactam, imipenem/relebactam, and cefiderocol have been developed and recently released in the market [13]. Thanks to their well-known tolerability and safety profile, β-lactam antibiotics are one of the most commonly used classes of antibiotics in the adult and pediatric population [14]. However, infants and neonates are not usually included in clinical development programs of new antibiotics, leading to a high usage of off-label and unlicensed prescriptions [15,16]. As a direct consequence, pharmacokinetic (PK) data derived from adult studies are often used to determine pediatric doses. Since multiple PK parameters are affected by the rapidly mutating physiologic and biochemical processes occurring in growing children, this practice can lead to inappropriate or unsafe dosing regimens [17,18]. This is even more evident for special populations such as patients with cystic fibrosis (CF), which may have altered PK parameters due to the larger volume of distribution and increased total body clearance of β-lactams leading to a smaller AUC, and a shorter elimination half-life [19,20]. Critically ill patients may also present marked changes in their PK parameters as a consequence of the increased renal clearance due to severe renal impairment and systemic inflammatory response that can affect drug bioavailability and distribution [21,22].

β-lactam antibiotics show a time-dependent bactericidal activity, which is optimal when the time (T) that the free drug concentration remains above the minimum inhibitory concentration (MIC) during dosing intervals (ƒT > MIC) is at least 40–70% of the total time of exposure [23]. However, in cases of critically ill patients or when susceptibility data are pending, a more aggressive target (up to 100% ƒT > 4–5 × MIC) has shown to result in better outcomes [24,25]. The PK variability in the pediatric population requires an extended use of therapeutic drug monitoring (TDM) in order to attain the pharmacodynamic (PD) targets and avoid drug-related toxicity [26].

In this narrative review, we analyzed the available pre-clinical, clinical, and pharmacological evidence regarding the use of the novel beta lactam antibiotics in the pediatric population in the effort of helping pediatricians to choose the most appropriate dosing regimen. 

## 2. Materials and Methods

We searched PubMed, Medline, and Embase with the following Mesh terms: “beta lactam” OR “carbapenem” OR “cephalosporin” OR “ceftolozane” OR “tazobactam” OR “ceftazidime” OR “avibactam” OR “meropenem” OR “vaborbactam” OR “imipenem” OR “relebactam” OR “cilastatin” OR “cefiderocol” AND “Gram negative” OR “drug” OR “resistant” OR “MDR” OR “bacteria” AND “pediatric” OR “pediatric” OR “child” OR “children”. Only English-written manuscripts published from January 2012 to April 2023 were considered. We also looked in the reference lists of the selected articles to retrieve further works and in the product information from the EMA (from 1995 periodically updated www.ema.eu, accessed on 29 May 2023) and FDA websites (www.accessdata.fda.gov, accessed on 29 May 2023).

## 3. Ceftolozane/Tazobactam

Ceftolozane/tazobactam is a combination of a novel broad-spectrum fifth-generation cephalosporine with a well-established β-lactamase inhibitor [27]. Ceftolozane/tazobactam has shown excellent activity against multidrug-resistant (MDR) *P. aeruginosa* [28,29] and extended spectrum beta-lactamase (ESBL)-producing Gram-negative bacteria [30].

Ceftolozane is structurally similar to ceftazidime but has a greater affinity to penicillin-binding protein 1b, 1c, 2, and 3 [31] and a reduced affinity for AmpC β-lactamase, and it is a weak substrate for efflux pumps [32,33,34]. Ceftolozane/tazobactam has been approved in adult patients for the treatment of complicated intra-abdominal infections (cIAI), complicated urinary tract infections (cUTIs) including pyelonephritis, and hospital-acquired pneumonia (HAP) including ventilator-associated pneumonia (VAP). Recently, the FDA and the EMA have approved the use of ceftolozane/tazobactam for pediatric patients with the exception of HAP and VAP [35,36]. The recommended dosage for pediatric patients is 20 mg/kg of ceftolozane and 10 mg/kg of tazobactam up to a maximum dose of 1 g of ceftolozane and 0.5 g of tazobactam infused over 1 h every 8 h [35,36].

The activity of ceftolozane/tazobactam against *P. aeruginosa* strains (including carbapenem resistant) has been extensively described in isolates from adults and children without CF [37]. Global surveillance studies analyzing isolates from four continents have shown that ceftolozane/tazobactam was the antimicrobial with the highest activity against *P. aeruginosa* after colistin [38,39]. Furthermore, ceftolozane/tazobactam has been demonstrated to retain activity even in the case of isolates resistant to ceftazidime/avibactam [40]. *P. aeruginosa* isolates from patients with CF can show a mucoid phenotype, characterized by the overproduction of exopolysaccharide alginate, expressing enhanced resistance to multiple antibiotics [41]. Most studies tested in vitro susceptibility to ceftolozane/tazobactam of *P. aeruginosa* strains collected from children with CF, indicating that ceftolozane/tazobactam is the most active β-lactam antibiotic against *P. aeruginosa* [42,43,44]. However, a recently published study including cefiderocol among the antibiotics tested against *P. aeruginosa* isolates collected even before the introduction of the novel β-lactams on the market showed 49% resistance to ceftolozane/tazobactam against 30% resistance to cefiderocol [45]. The acquisition of resistance to ceftolozane/tazobactam has been mostly associated with the overexpression of AmpC, which may become particularly alarming because it is associated with cross-resistance with ceftazidime/avibactam [46,47,48]. An additional study on 28 patients treated with ceftolozane-tazobactam and ceftazidime-avibactam showed 86% resistance in the patients’ isolates previously susceptible to the agent [49].

PK studies have been conducted to assess safety, tolerability, and drug exposure levels in pediatric participants showing comparable values with those observed in adults [50,51]. In a phase 1 multicenter study were enrolled 37 patients (aged from 7 days postnatal to 18 years) with proven or suspected Gram-negative infections or receiving perioperative prophylaxis [50]. Ceftolozane PK was generally comparable among children older than 3 months, while the clearance of both ceftolozane and tazobactam was lower in patients <3 months of age, and the volume of distribution was slightly higher. This difference is consistent with the immature renal function in this age group since both products are cleared by the kidney. The ceftolozane-tazobactam dosages were 30/15 mg/kg and 20/10 mg/kg, respectively, in children aged between 3 months and 7 years and in neonates and infants <3 months of age. It achieved comparable drug exposures with those previously observed in healthy adult volunteers. The PK data from the abovementioned study and from an additional 12 adults included in other studies were integrated in one population pharmacokinetic analysis [52]. This analysis, conducted to guide the dose selection for two subsequent phase 2 clinical trials for the treatment of cIAI and cUTI in children, showed that renal function significantly affects ceftolozane/tazobactam clearance and that body weight influences both ceftolozane/tazobactam clearance and volume of distribution. The activity of ceftolozane/tazobactam against *P. aeruginosa* makes it a valuable molecule for patients with CF [53]. PK in CF patients results were altered because of the larger volume of distribution, increased total body clearance of β-lactams, smaller AUC, and shorter elimination half-life [19,20]. These factors may lead to lower antibiotic concentrations, needing increased dosages to achieve adequate plasma and epithelial lining fluid concentrations. However, studies conducted to explore and assess the PK of ceftolozane/tazobactam in patients with CF have failed to demonstrate clinically significant differences in weight-normalized plasma concentrations of ceftolozane/tazobactam evaluated in children with or without CF [54,55]. The ceftolozane/tazobactam PK in critically ill pediatric patients has been studied in only one small cohort of three patients, presenting with normal renal function, acute kidney injury (AKI), and AKI with the necessity of continuous renal replacement therapy (CRRT) [56]. These findings suggested modifying the dosing regimens for ceftolozane of 35 mg/kg every 8 h in patients with normal renal function, 10 mg/kg for children with severe AKI, and 30 mg/kg for patients undergoing CRRT. 

At the moment, in vivo evidence related to the use of ceftolozane/tazobactam in children and adolescents is still limited (Table 1).

Multiple case reports have described the use of ceftolozane/tazobactam in children with serious baseline co-morbidities and bloodstream infection [57], pulmonary exacerbation [58], endocarditis [59], severe pneumonia [60], and vascular graft infection [61]. Molloy et al. [62] described a cohort of 13 pediatric patients with MDR-*P. aeruginosa* infections. All patients achieved a clinical cure except for one, who died due to the underlying disease and cardiovascular failure unrelated to infection. The ceftolozane/tazobactam treatment was well tolerated except for one case of high levels of transaminases and another one of neutropenia, both resolved with a reduction of the dose used. In two patients receiving prolonged or repeated courses of ceftolozane/tazobactam, *P. aeruginosa* developed resistance to ceftolozane/tazobactam (MIC > 256 mg/L). The dosing strategies for most patients consisted of 20 mg/kg every 8 h of ceftolozane, which was increased in cases of serious respiratory infections to 30–40 mg/kg every 8 h, up to a maximum of 2 g for doses. Two phase 2 randomized, double-blind studies were conducted to permit the approval of ceftolozane/tazobactam use in the pediatric population [63,64]. In both studies the selected doses were based on population pharmacokinetic modeling and simulations [50,52]. Jackson et al. [63] have assessed the use of ceftolozane/tazobactam in pediatric patients (<18 years) with cIAI, comparing ceftolozane/tazobactam plus metronidazole versus meropenem. A total of 91 patients were enrolled in the study, and 70 received the combination of ceftolozane/tazobactam plus metronidazole. The most common Gram-negative pathogens were *E. coli*, *P. aeruginosa*, and *Bacteroides fragilis*, and all isolates were susceptible to ceftolozane/tazobactam and meropenem. The rates of clinical cure evaluated at the end of treatment were 80% for C/T plus metronidazole and 95.2% for meropenem; the overall per-participant microbiologic success rates at the test of the cure visit were high (>84%) and comparable between treatment groups. Ceftolozane/tazobactam plus metronidazole was well tolerated by the study population, and the safety profile was similar to that described in adults. Roilides et al. [64] have assessed the safety and efficacy of ceftolozane/tazobactam compared with meropenem for the treatment of cUTI, including pyelonephritis, in a total of 95 neonatal and pediatric patients. Pyelonephritis was the most common clinical presentation, and *Escherichia coli* was the most common pathogen. The clinical cure rate at the end of the treatment was 94.4% for C/T and 100% for meropenem. Among the 71 patients treated with ceftolozane/tazobactam, the safety profile was comparable to meropenem and to the previously reported safety profile for ceftolozane/tazobactam in adults with cUTI. At the moment, a phase 1 clinical trial is being conducted, aimed to evaluate the safety, the tolerability, and the PK of ceftolozane/tazobactam in pediatric patients (from birth to 18 years of life) with HAP/VAP [65]. The study proposes an increased dose of 60 mg/kg (40 mg/kg ceftolozane, 20 mg/kg tazobactam) administered intravenously every 8 h as a 1 h infusion (Table 2).

**Table 1 microorganisms-11-01798-t001:** Clinical characteristics, management, and outcomes of ceftolozane/tazobactam in infections due to MDR Gram-negative bacteria in the pediatric population.

Study	Study Design	Population	Infection	C/T MIC (μg/mL)	C/T Dosage and Duration	TDM (μg/mL)	AEs	Outcomes
**Roilides et al.** [64]	Phase 2, randomized, double-blind study C/T compared with meropenem	mMITT: from birth (>32 weeks gestational age and ≥7 days postnatal) to <18 years of age (C/T group *n* = 71 meropenem *n* = 24)	cUTI Most common pathogens: *E. coli* (74.6%) *K. pneumonia* (8.5%) *P. aeruginosa* (7%)	NA	From birth to <12 years 20/10 mg/kg q8h in 1 h infusion From 12 to <18 years 1.5 g q8h in 1 h infusion	NA	≥1 AEs 59.0% (59/100) C/T 60.6% (20/33) meropenem	Clinical cure rates EOT (mMITT population): 94.4% (67/71) C/T 100% (24/24) meropenem
**Jackson et al.** [63]	Phase 2, randomized, double-blind study C/T + MTZ compared with meropenem	MITT: from birth (>32 weeks gestational age and ≥7 days postnatal) to <18 years of age (C/T *n* = 70 meropenem *n* = 21)	Presumed or documented cIAI Most common diagnosis: complicated appendicitis (91.4%) Most common pathogens: *E. coli* (67.1%) *P. aeruginosa* (27.1%) *Bacteroides fragilis* (18.6%)	NA	From birth to <12 years 20/10 mg/kg in 1 h infusion + IV MTZ 10–15 mg/kg From 12 to <18 years 1,5 g in 1 h infusion + IV MTZ 10–15 mg/kg	NA	No AEs leading to death, drug-related serious AEs‚ or discontinuations due to drug-related AEs or serious AEs	Clinical cure rates EOT (MITT population): 80% (vs. 95.2% meropenem)
**Molloy et al.** [62]	Case series	Patients aged 0.25–19 years (*n* = 13)	MDR-*P. aeruginosa* infections Pneumonia (*n* = 8) CF exacerbation (*n* = 3) IAI (*n* = 2) Osteomyelitis (*n* = 1)	0.06 (*n* = 1) 0.5 (*n* = 2) 2 (*n* = 6) 2 (*n* = 3) 4 (*n* = 1)	−20/10 mg/kg q8h −30/15–40/20 mg/kg (max 2/1 g) q8h for serious respiratory infections	NA	Elevation of transaminitis (*n* = 1) Neutropenia (*n* = 1)	Clinical resolution (*n* = 12/13)
**Perruccio et al.** [66]	Retrospective, observational study	Children with hematological malignancies (*n* = 4) (subgroup patient characteristics NA)	MDR Gram-negative infections (subgroup microbiological characteristics NA)	NA	1.5 g q8h for a median of 20 days (range: 14–20) One patient who weighed <10 kg: 200/100 mg q8h	NA	Subgroup description of AEs NA	Subgroup analysis NA
**Aitken et al.** [57]	Case report	9-year-old male patient with acute myeloid leukemia	Two episodes of MDR-*P. aeruginosa* BSI	6 (first episode) 8 (second episode)	50/25 mg/kg q8h over a 3 h infusion (first treatment) + tobramycin and ciprofloxacin for 3 weeks 40/20 mg/kg q6h over a 3 h infusion + tobramycin and ciprofloxacin for 3 weeks	C_min_ (C): 5.2 C_max_ (C): 74.1 C_min_ (C): 18.1 C_max_ (C): 54.3	None	Clinical and microbiological resolution
**Ang et al.** [58]	Case report	14-year-old female with cystic fibrosis	*P. aeruginosa* pulmonary exacerbation (two strains: mucoid and nonmucoid)	0.5 (mucoid) 1 (nonmucoid)	C/T 1.5 g q8h in 1 h infusion for 14 days	C_max_ (C): 94.1 C_min_ (C): 1.2 C_max_ (T): 12.1 C_min_(T): 0.04	Elevation of transaminasis	Clinical resolution
**Martín-Cazaña et al.** [59]	Case report	5-year-old male with complex congenital heart disease	MDR-*P. aeruginosa* endocarditis	2	50/25 mg/kg q8h over 3 h infusion + tobramycin for 45 days	C_max_ (C): 72.9 C_min_ (C): 2.6	None	Clinical and microbiological resolution
**Zikri and El Masri** [60]	Case report	14-year-old female with combined immunodeficiency syndrome	MDR-*P. aeruginosa* pneumonia	3	1.5 g q8h + amikacin and colistin	NA	None	Clinical resolution
**Dinh et al.** [61]	Case report	3-year-old male with liver transplant	XDR-*P. aeruginosa* vascular graft infection	NA	1.5/0.75 g/day for 57 days + colistin	NA	*Clostridioides difficile* infection	Clinical and microbiological failure

BSI—bloodstream infection; C/T—ceftolozane/tazobactam; C—ceftolozane; T—tazobactam; q8h—every 8 h; AEs—adverse effects; TDM—therapeutic drug monitoring; CF—cystic fibrosis; IAI—intra-abdominal infection; MIC—minimum inhibitor concentration; NA—not available; C_max_—peak concentration; C_min_—trough concentration; mMITT—microbiologic modified intent to treat; MITT—modified intent to treat; EOT—end of treatment; MTZ—metronidazole.

### Summary

Although in pediatric patients the use of ceftolozane/tazobactam has proved to be safe and effective, the small size of patients treated suggests caution with this therapeutic choice. Ceftolozane/tazobactam has not yet been licensed for the treatment of pulmonary infections, where it proved to be successful in the adult population [67], including adults with CF [68,69]. This lack of data and approval collides with the increasingly urgent need for novel agents for the treatment of MDR Gram-negative bacteria. At the same time, more PK/PD studies are needed to guide dosing strategies in children, especially in selected populations such as critically ill patients with renal impairment. Further PK/PD studies are needed in order to approve the use of ceftolozane/tazobactam for HAP/VAP in children and to confirm the need of higher doses such as for the treatment of pneumonia in adults.

## 4. Ceftazidime/Avibactam

Ceftazidime/avibactam is the combination of a well-known third-generation cephalosporine, ceftazidime and avibactam, a novel synthetic β-lactamase inhibitor capable of neutralizing the activity of ESBLs, AmpC β-lactamases, and both KPC and OXA-48 carbapenemases but unable against organisms producing metallo-β-lactamases and *Acinetobacter* spp. [27,70]. When tested against carbapenem-resistant Enterobacterales (CRE) strains, 80% of the isolates showed in vitro susceptibility [71]. As for *P. aeruginosa,* the majority (about 90%) of the isolates collected from four continents were susceptible [72], and also when tested against CR-*P. aeruginosa*, ceftazidime/avibactam retained activity in 76% of the isolates [73]. The emergence of resistance to ceftazidime/avibactam has been increasingly described and appears related to mutations in the omega loop of the KPC enzyme leading to enhanced ceftazidime hydrolysis in KPC-2 and KPC-3 producing isolates of Enterobacterales [74,75,76,77,78]. Of concern is the emergence of resistance to ceftazidime/avibactam during treatment, which warrants the attentive vigilance of resistance development during treatment and appropriate infection and prevention control measures [79,80,81,82]. Resistance to ceftazidime/avibactam is further a matter of concern due to the conferred cross-resistance to the novel siderophore–cephalosporin conjugate, cefiderocol [83].

Ceftazidime/avibactam has been approved in Europe and in the US for pediatric patients aged 3 months and older for the treatment of cIAI, cUTI including pyelonephritis, and HAP including VAP [84,85]. In Europe it has also been approved to treat infections due to aerobic Gram-negative bacteria with limited treatment options [85]. The recommended dosage for patients, aged between 6 months and 18 years, is 50 mg/kg of ceftazidime and 12.5 mg/kg of avibactam up to a maximum dose of 2 g ceftazidime and 0.5 g of avibactam infused over 2 h every 8 h. In cases of patients with ages between 3 months and 6 months the recommended dosage is 40 mg/kg of ceftazidime and 10 mg/kg of avibactam [84,85].

In the pediatric population the activity of ceftazidime/avibactam against MDR Gram-negative isolates collected from children has shown high levels of susceptibility. In one pediatric hospital in the U.S. the reported rate of in vitro susceptibility of Gram-negative isolates of *Enterobacterales* and *P. aeruginosa* was superior at 99% for both pathogens [86]. The bacterial isolates collected from pediatric patients with UTI and IAI at 70 medical centers resulted in 100% susceptibility of Enterobacterales strains, while *P. aeruginosa* strains were 96.2% and 100% susceptible in the cases of UTI and IAI, respectively [87]. However, a recent study analyzed 66 isolates from CF pediatric patients from a German hospital reported that up to 53% of the isolates were resistant to ceftazidime/avibactam [45]. This finding raises concern, although it may be limited in this specific population.

The PK profile, safety, and tolerability of a single dose of ceftazidime/avibactam in 32 children (≥3 months to <18 years) hospitalized for a suspected or confirmed infection were evaluated during a phase 1 study [88]. Patients, divided in four groups according to the age, received 2/0.5 g and 50/12.5 mg/kg over 2 h if >40 kg or <40 kg, respectively. PK population modeling was used to describe the PK characteristics across all age groups. The mean plasma concentrations were homogeneous in all four groups for both drugs, and the PK profiles were comparable to those previously observed in the adult population. The study did not identify any safety concern providing sufficient data to guide dosing strategies for the subsequent phase 2 studies in pediatric patients. This dosage regimen was also supported by a population PK study that pooled PK data extrapolated from the previous phase 1 study and the phase 2 pediatric trials in children affected by cUTI and cIAI together with data from phase 1 and phase 3 trials in adults [89]. The combined dataset was used to update the PK model and run simulations demonstrating that the exposure and probability of target attainment in pediatric patients with normal or mildly impaired renal function were comparable to those observed in adults. Using a similar methodology, a subsequent PK population modeling study based on an extensive adult and pediatric database was conducted [90]. The results supported the recommended dose adjustments for pediatric patients ≥2 to <18 years old with moderate, severe, or very severe renal impairment or ESRD and cIAI, cUTI, or HAP/VAP and for those ≥3 months to <2 years old with moderate or severe renal impairment. To date, the clinical data on infants aged <3 months are very limited, and it is mainly used during off-label protocols and supported by the center’s experience. However, a phase 2 study evaluating the safety, pharmacokinetics, and tolerability of ceftazidime/avibactam in neonates and infants aged between 26 weeks of post-menstrual age and 3 months is ongoing [91] (Table 2).

Several case reports and case series have described the use of ceftazidime/avibactam for the treatment of MDR Gram-negative infections in children and infants [3,66,92,93,94,95,96,97,98] (Table 3).

Ren et al. [93] have recently described a case of a 4-year-old girl with post-neurosurgical meningitis and abscess caused by ESBL-producing *E. coli* successfully treated with ceftazidime/avibactam. Of note, the TDM of ceftazidime/avibactam was performed in the serum and CSF achieving CSF/blood barrier penetrations of approximately 27.3 ± 0.4% and 40.5 ± 7.7% for ceftazidime and avibactam, respectively. An additional case report recently described the successful treatment of a boy with a ventriculoperitoneal shunt infection due to MDR *P. aeruginosa* with ceftazidime/avibactam and intraventricular colistin [94], providing promising evidence for the use of ceftazidime/avibactam in CNS infections and warranting further studies. Moreover, a recently published multicenter retrospective analysis described 25 pediatric patients with hematologic malignancies and febrile neutropenia, treated with ceftazidime/avibactam (*n* = 21) and ceftolozane/tazobactam (*n* = 4) as empiric, first-line, or second-line targeted therapy [92]. Even in this category of deeply immunosuppressed patients, both β-lactams proved to be safe and effective, achieving infection resolution in more than 90% of patients.

As part of the global clinical development program to support the extension of the indication of ceftazidime/avibactam treatment, two phase 2 randomized, single-blind trials have been conducted in pediatric patients with cUTI and cIAI and dosing regimens based on the previously discussed phase 1 population pharmacokinetic study [88,99]. In both studies, the children were randomized 3:1 and divided in four cohorts to receive ceftazidime/avibactam alone in cUTI or plus metronidazole in cIAI and compared with cefepime and meropenem, respectively. In both studies, favorable clinical outcomes (>90%) were achieved with all drugs tested, while the safety profile and tolerability were consistent with those reported in adults. Also, the PK analysis observed mean plasma concentrations to be homogenous in all cohorts.

The combination of ceftazidime/avibactam plus aztreonam has been proposed as a potential treatment option to treat infections due to metallo-β-lactamase-producing Gram-negative bacteria and overcome resistance. In fact, aztreonam retains activity against metallo-β-lactamases but needs to be supported by other agents such as ESBLs and KPC to avoid its inactivation [100,101,102]. Pediatric experience on this combination is limited to a few case reports [103,104]. In particular, Cowart et al. [104] described the case of an 11-year-old female patient with CF and persistent pulmonary exacerbations caused by *Stenotrophomonas maltophila* treated with continuous infusion of aztreonam and ceftazidime/avibactam. TDM was also performed to evaluate the drug exposure and probability of target attainment. Therapy with ceftazidime/avibactam infused over 2 h, aztreonam over 3 h, and minocycline was started. Blood sampling showed suboptimal concentrations for aztreonam and ceftazidime; a switch to a continuous infusion regimen of both drugs optimized the percentage of time when the free drug remained above the minimum inhibitory concentration (*f*T > MIC). As this case shows, TDM and continuous infusion may be used as real-life improving tools in resistant isolates.

### Summary

According to the available data ceftazidime/avibactam was demonstrated to be a safe and effective treatment option in pediatric patients with cUTI, cIAI, and HAP/VAP caused by MDR Gram-negative bacteria. Alarmingly, resistance to this molecule has been increasingly described worldwide. Cross-resistance with cefiderocol and the development of drug resistance during treatment are concerning and need careful microbiological monitoring and surveillance. In our opinion, in order to preserve this molecule action, it should not be used in carbapenem-sparing strategies but reserved against KPC and OXA-48 producers only [105,106]. Although promising, the use of ceftazidime/avibactam in different infection sites (such as the CNS) is not sustained by sufficient evidences and should be restricted to cases with very limited treatment options.

## 5. Meropenem/Vaborbactam

Meropenem/vaborbactam is the first antimicrobial combination of a novel, cyclic, boronic acid-base β-lactamases inhibitor with a carbapenem backbone [107]. Meropenem is a broad-spectrum carbapenem, with excellent activity against ESBLs and strong safety profile and tolerability that contributed to its extensive use for the treatment of severe Gram-negative bacterial infections in the last two decades. Vaborbactam inhibits class A and class C β-lactamases with particularly potent activity against KPC. Vaborbactam was paired with meropenem to restore its activity against KPC-producing Enterobacterales but not against metallo-β-lactamases or OXA-48-like enzymes [108,109]. In the case of *P. aeruginosa* and *A. baumanni*, the activity of meropenem/vaborbactam is similar to that of meropenem alone since the resistance is mediated by mechanisms that are not contrasted by vaborbactam, such as cell membrane impermeability, increased activity of efflux systems, and production of class B or class D β-lactamases [110,111].

Meropenem/vaborbactam has been approved in Europe and in the US for the treatment of cUTI, cIAI, and HAP including VAP in adult patients (>18 years). The recommended dosing regimen consists of 2 g/2 g every 8 h, as a 3 h extended infusion, for patients with normal renal function [112,113]. These indications were based on a phase 3, randomized, open-label trial (TANGO II) that investigated adult patients with cUTIs, HAP/VAP, bacteremia, or cIAIs due to confirmed or suspected CRE, 63% of which were KPC-producing. Meropenem/vaborbactam was compared with the best available therapy, showing superior results, especially in immunocompromised patients [114].

To date, PK safety and efficacy data about meropenem/vaborbactam in children are not available, and pediatric experiences are limited to case reports. Harnetty et al. [115] described the case of a 4-year-old child with several gastro-pulmonary co-morbidities and bloodstream infection due to KPC-producing *K. pneumoniae*. Meropenem/vaborbactam was started at a dose of 40 mg/kg every 6 h infused over 3 h, based on pharmacokinetic data of meropenem in critically ill children. The dosing strategy allowed a target attainment of 100% of meropenem serum concentrations above the MIC for at least 40% of the time between dosing intervals. Patients were successfully treated for 14 days with clearance of the bacteremia, and no safety concerns were reported. The use of meropenem/vaborbactam co-administered with cefiderocol and bacteriophage therapy has been reported in a 10-year-old girl with CF with a pan-drug-resistant (PDR) *Achromobacter* spp [116] infection in two separate admissions. Clinical improvement and a microbiological cure were obtained, and the combination proved to be well tolerated and safe. At the moment, an open-label phase 1 trial is ongoing to evaluate the dosage, pharmacokinetics, safety, and tolerability of a single “dose infusion” of meropenem-vaborbactam in pediatric patients, from birth to <18 years of age with serious bacterial infections (TANGOKIDS) [117] (Table 2).

### Summary

Meropenem/vaborbactam is a promising treatment option in infants and children with MDR Gram-negative infections, especially KPC-producing Enterobacterales. Although pediatric data are solely limited to case reports, results from adult studies show high efficacy rates, safety and tolerability. Furthermore, the low propensity of the molecule to induce resistance also with extended infusion-dosing regimen, which allows the attainment of aggressive PK/PD targets, could be of great value for pediatric patients in intensive care settings due to critically ill. However, due to the lack of clinical data in the pediatric population, while waiting for more detailed information on the molecule from the ongoing clinical trial, other antibiotics with approved indications should be preferred (e.g., ceftazidime/avibactam), and the use of meropenem/vaborbactam should be restricted to select cases with very limited treatment options.

## 6. Imipenem/Cilastatin/Relebactam

Relebactam is a bicyclic diazabicyclooctane, β-lactamase inhibitor that is structurally similar to avibactam [118,119]. It inhibits Ambler classes A KPC and C AmpC β-lactamases [120,121], but it is not active against class B metallo-β-lactamases (including IMP, VIM, and NDM) or class D OXA-48 [122]. In addition to class B and class D β-lactamases, resistance to imipenem/cilastatin/relebactam in Enterobacterales is also associated with OmpK35 and OmpK36 nonfunctional mutations [123], while *P. aeruginosa*’s resistance to imipenem is mediated by AmpC overproduction or OprD porin’s decreased expression [121,124]. The addition of relebactam increases activity against CR-*P. aeruginosa*, being a potent inhibitor of *P. aeruginosa* AmpC and of other Pseudomonas-derived cephalosporinases [121,124,125]. However, the addition of relebactam does not improve imipenem activity against *A. baumanni* and *S. maltophilia* [126,127]. Imipenem is a carbapenem susceptible to degradation by the enzyme dehydropeptidase-1 (DHP-1) located in renal tubules and requires co-administration with a DHP-1 inhibitor, such as cilastatin, which has no antibacterial activity [128]. For reasons of clarity, in this review we will refer to imipenem/cilastatin just as imipenem, unless otherwise specified.

Imipenem/relebactam susceptibility against Enterobacterales and *P. aeruginosa* isolates was tested in a global surveillance program (SMART) in 2015. The susceptibility was >90% against *K. pneumoniae*, *P. aeruginosa*, and *Enterobacter* spp. and 74.1%, 80.5%, and 100% in imipenem-resistant isolates. respectively [126]. The data from SMART surveillance in 2015–2017 evaluating imipenem/relebactam in vitro activity in isolates collected from ICU patients with low respiratory tract infections report *P. aeruginosa* susceptibility of 92.2%, including 77.2% of imipenem-resistant isolates [129].

The FDA and the EMA approved imipenem/relebactam in 2020 to treat the HAP, including VAP, in adult patients (>18 years) at the approved dosage of 1.25 g (imipenem 500 mg, cilastatin 500 mg, relebactam 250 mg) every 6 h infused in 30 min. In addition, it is also indicated in the US for cUTI and cIAI with limited treatment options, while in Europe it is indicated for HAP/VAP-associated bacteremia and infections due to aerobic Gram-negative organisms [130,131]. These indications are mainly based on two randomized, controlled, and comparative phase 3 clinical trials on the imipenem/relebactam use in adults. The first study assessed the efficacy, safety, and tolerability of imipenem/relebactam compared with imipenem plus colistin in imipenem-nonsusceptible bacterial infections, resulting in an overall favorable response in 71% and 70% of patients with a greatly reduced nephrotoxicity in the imipenem/relebactam arm (10% vs. 56% *p* = 0.002) [132]. The second trial compared imipenem/relebactam with piperacillin/tazobactam in 537 patients (66.1% in the ICU), showing noninferiority to the comparator and favorable clinical outcomes in the subgroup of patients with an APACHE II score >15 [133].

The PK, safety, and tolerability of imipenem/relabactam in the pediatric population have been evaluated in an open-label phase 1 single-dose clinical trial [134]. Initial dosing regimens of 15/7.5 mg/kg in patients from 2 years to <18 years and 10/5 mg/kg in patients from birth to 2 years were subsequently modified after an interim review in 500/250 mg from 12 to <18 years and 15/7.5 mg/kg from birth to 12 years. The %ƒT > MIC for imipenem exceeded the objective of 30%, ranging from 55 to 94% across all pediatric age cohorts and proving to be well tolerated. At the moment, an ongoing phase 2/3 clinical trial aims to compare imipenem/relebactam with an active control in 140 pediatric patients with HAP/VAP, cUTI, and cIAI [135].

### Summary

Imipenem/relebactam resulted as being safe and effective in the adult population and should be considered in the treatment of HAP/VAP, cUTI, and cIAI. To date, the microbiological and clinical data on resistance selection during treatment are too limited to express a preference toward this molecule over others with the same spectrum of activity. Regarding the pediatric population, the available clinical studies are scarce and mainly are about the PK of this antimicrobial agent. For these reasons, imipenem/relebactam should not be considered as a treatment option in children if other active agents are available.

## 7. Cefiderocol

Cefiderocol is a novel siderophore cephalosporin, structurally similar to cefepime and ceftazidime, with a unique mechanism of inhibiting the cell wall synthesis of Gram-negative bacteria. Thanks to its siderophore-like properties, cefiderocol is transported in the cellular periplasmic space via active ferric iron transporters, where it dissociates from the iron and binds to penicillin-binding proteins, causing the inhibition of peptidoglycan cell wall synthesis [136,137]. This mechanism of action confers to cefiderocol high stability to all four Ambler classes of β-lactamases and carbapenemases produced by *Enterobacterales*, including ESBLs, AmpC, KPC, NDM, VIM, IMP, and OXA-48 [136]. Cefidercol also exhibits potent activity against *A. baumanni*, *P. aeruginosa*, *S. maltophilia*, and *Burkholderia* spp. Resistance to cefiderocol in *A. baumanni* has been associated with reduced expression of the siderophore receptor gene pirA and PBP3 mutations [138]. In a recent study on 66 *P. aeurginosa* isolates collected from pediatric and adolescent CF patients, 30% of the isolates were resistant to cefiderocol, considerably less than those resistant to ceftazidime avibactam (49%) and ceftolozane/tazobactam (53%) [45].

FDA approved cefiderocol for the treatment of cUTI and HAP/VAP, while the EMA indication regards the treatment of infections due to aerobic Gram-negative organisms in adults with limited treatment options [139,140]. The prescribed dosing regimen in adults is 2 g every 8 h as an extended infusion over 3 h. These indications were in accordance with the results obtained during a phase 2 trial that demonstrated the noninferiority to imipenem-cilastatin for the treatment of cUTI in adult patients [141]. Subsequently, in two phase 3 studies, cefiderocol was proven to be noninferior to the best available therapy and to high-dose extended infusion meropenem for Gram-negative bacterial infections, regardless of species or source of infection and nosocomial pneumonia, respectively [142,143]. As for other β-lactams, %fT > MIC is the main PK/PD parameter that correlates with efficacy for cefiderocol [144]. A study on thigh and lung infection models shows that the mean %fT > MIC required for a log10 reduction in colony forming units for carbapenem-resistant Gram-negative bacteria was 64.4% for Enterobacterales, 70.3% for *P. aeruginosa*, 88.1% for *A. baumannii*, and 53.9% for *S. maltophilia* [145]. According to population PK/PD models, the extended infusion regimen is expected to reach these exposures [146].

At the moment, cefiderocol is not approved for the pediatric population, and the clinical data are based on a few case reports. Alamarat et al. [96] described the use of long-term cefidercol (14 weeks) in the treatment of chronic osteomyelitis due to extensively drug-resistant (XDR) *P. aeruginosa* NDM-1 metallo-β-lactamase producer in a 15 year-old girl, with an apparent cure and an avoided amputation. Cefiderocol was well tolerated, but intermittent episodes of decreased white cell counts with spontaneous resolution were reported. Monari et al. [147] reported the case of a preterm newborn with bloodstream infection due to a VIM-producing *K. pneumoniae* treated successfully for 9 days with cefiderocol given at 60 mg/kg as a loading dose followed by 40 mg/kg every 8 h in a 3–4 h extended infusion regimen. A recent case of *S. maltophilia* bacteremia in an infant initially treated with cotrimoxazole and subsequently switched to cefiderocol given at 40 mg/kg/dose every 8 h infused over 1 h resulted in bacterial clearance in 24 h [148].

The PK, safety, and tolerability of cefiderocol in hospitalized pediatric patients (3 months to <18 years) have been assessed in a recently completed clinical trial, but the results have not been published yet [149]. There is a phase 2 trial consisting of a first part, aimed at assessing the PK of cefiderocol in hospitalized children, and a second randomized part, in which the cefiderocol plus standard of care will be compared with standard of care alone in pediatric patients with cUTI, HAP, or VAP (Table 2) [150]. In both studies, the proposed doses are 60 mg/kg every 8 h for weight <34 kg and 2 g every 8 h for those weighing ≥34 kg. All participants received cefiderocol via a 3 h extended infusion regimen.

### Summary

Cefiderocol with its wide spectrum of activity that comprises all types of β-lactamases has a prominent position among the new β-lactam molecules released recently. However, to now there is an evident lack of clinical information, particularly regarding its use, on pediatric population. In order to preserve this antibiotic, cefiderocol should be reserved for the treatment of infections due to metallo-β-lactamase-producing Enterobacterales, MDR-*P. aeruginosa*, and other MDR Gram-negative bacteria when ceftlozane/tazobactam, ceftazidime/avibactam, meropenem/vaborbactam and imipenem/relebactam are unsusceptible or unavailable.

## 8. Discussion

This review offers an overview on the five antibiotics recently approved for the treatment of MDR Gram-negative infections, focusing on pediatric aspects. To date, the FDA and EMA have approved for pediatric patients only two of the antibiotics described above: ceftolozane/tazobactam and ceftazidime/avibactam [35,36,82,83]. Therefore, the approved armamentarium for children against MDR Gram-negative infections is still limited and often based on the experience extrapolated by case series or single case reports.

Local epidemiology of multidrug-resistant microorganisms plays a role in choosing these new antibiotics. Few reports are available in the literature about the epidemiology of CRE infection in children, with differences among countries [3,11,151,152]. Historically, the antibiotic options for the treatment of carbapenem-resistant bacteria included polymyxins, tigecycline, and aminoglycosides, often used in combination therapies. However, the limitations due to the side effects of these agents (i.e., nephrotoxicity) are well known and often have a negative impact on patient outcomes. Thus, further studies are needed to compare the differences in efficacy and safety between “old” antibiotics and “new” agents. In 2023, Tripiciano et al. [153] conducted a study on 42 pediatric patients affected by infections due to carbapenemase-producing microorganisms, comparing the outcomes in patients treated with new-generation cephalosporins (N-CEF) with those of patients treated with colistin-containing regimens (COLI). The statistical analysis showed that the N-CEF-containing treatment regimen was statistically associated with complete recovery (*p* = 0.04) and was noninferior to the COLI-containing treatment regimens. The treatment of CRE infections in children is complex. Therapeutic decisions require expert consultation and a personalized approach, often based on adult data, given the dearth of pediatric studies. The meropenem MIC of the infecting isolate, the type of carbapenemase-produced illness severity, and the source of the infection should be considered when selecting antibiotic therapy. Finally, while the treatment recommendations contained herein reflect the currently available data, treatment paradigms are likely to evolve over time as agents in the antibiotic pipeline become available and pediatric experience with available agents grows [154]. Regardless of the selected therapy, the fundamental concepts of effective antimicrobial treatment in critically ill children remain: the proper culture and molecular techniques for a rapid identification of the pathogen taking into account “the local epidemiology”, the timely initiation of therapy selecting agents with a high likelihood of susceptibility, and sufficient penetration in the site of infection, monitoring the adequate doses and intervals to enhance bactericidal activity [155].

To achieve the proper use of these new antibiotics in pediatrics, in addition to safety and efficacy trials, detailed PK studies should be implemented in order to determine the best dose regimen among specific age populations.

This will allow clinicians to administer the appropriate dosage, avoiding the onset of new resistances and thus preserving the few new weapons at our disposal. Furthermore, the regulatory authorities should commit to PK studies in pediatric populations, even for old drugs that are reappearing in wards as a viable option against MDR pathogens. In addition, pharmaceutical companies should also be strongly encouraged to engage in the pharmacological development of new antibiotics by including children in trials.

## 9. Conclusions

The right treatment of MDR infections in pediatrics, especially CRE infections, remains a challenge for pediatricians. Although the available data, mainly referring to adult studies in terms of efficacy, suggested high effectiveness and favorable safety profiles of the considered novel antimicrobials, more pediatric PK/PD studies are needed to address carefully the indications and safety profiles in all age groups of children. The collaboration between scientists, regulatory bodies, and healthcare professionals is important to improve the management of MDR infections in pediatric patients, also taking into account the “variability” between different patients in terms of clinical manifestations and genetic backgrounds but also considering the *local epidemiology*, the *center expertise in the field* (identification of correct pathogen), *the timely initiation of therapy*, and the *patient’s clinical condition*. All these factors could influence the efficacy of the treatment.

## Figures and Tables

**Table 2 microorganisms-11-01798-t002:** Ongoing clinical trials of the novel β-lactam antibiotics in the pediatric population.

β-Lactam Agent	Trials under Investigation				
	Study	Population	Investigated Dose	Status	Completion Date *
**Ceftolozane/tazobactam**	Safety and Pharmacokinetics of Ceftolozane/Tazobactam in Pediatric Participants With Nosocomial Pneumonia (MK-7625A-036) (NCT04223752)	Infants and children from birth to <18 years of age with nosocomial pneumonia	>12 to <18 years of age: 2/1 g over a 60 min <12 years of age: 40/20 mg/kg over a 60 min	Recruiting	September 2025
**Ceftazidime/avibactam**	Evaluation of Pharmacokinetics, Safety, and Tolerability of Ceftazidime-Avibactam in Neonates and Infants (NOOR) (NCT04126031)	Neonates and infants aged 26 weeks post-menstrual age to <3 months (participants enrolled *n* = 48)	NA	Terminated due to sponsor decision	NA
	Safety and Tolerability of Ceftazidime-Avibactam for Pediatric Patients With Suspected or Confirmed Infections (NCT01893346)	Children from 3 months of age to <18 years (participants enrolled *n* = 35)	>12 to <18 years: 2/0.5 g >6 to <12 years: 2/0.5 g (>40 kg), 50/12.5 mg/kg (<40 kg) >3 months to <6 years: 50/12.5 mg/kg	Completed	NA
**Meropenem/vaborbactam**	Dose-Finding, Pharmacokinetics, and Safety of Vabomere in Pediatric Subjects With Bacterial Infections (TANGOKIDS) (NCT02687906)	From birth to less than 18 years of age with serious bacterial infections	>3 months to <12 years: 60 m/kg >6 years to <18 years: 40 mg/kg Birth to <3 months: TBD 2 to <12 years: <35 kg 80 mg/kg	Recruiting	December 2023
**Imipenem/relebactam**	Safety, Tolerability, Efficacy and Pharmacokinetics of Imipenem/Cilastatin/Relebactam (MK-7655A) in Pediatric Participants With Gram-Negative Bacterial Infection (NCT03969901)	From birth to less than 18 years of age with confirmed or suspected Gram-negative bacterial infection	12 to <18 years: 500/250 mg q6h hours over 30 min 3 months to <2 years: 15/7.5 mg/kg, q6h over 30 min Birth to <3 month: 15/7.5 mg/kg q8h over 30 min	Recruiting	February 2024
	Imipenem/Cilastatin/Relebactam Pharmacokinetics, Safety, and Outcomes in Adults and Adolescents With Cystic Fibrosis	12 years and older	Adolescents: 15/7.5 mg/kg q6h over 30 min	Recruiting	December 2023
**Cefiderocol**	A Study to Assess the Safety, Tolerability, and Pharmacokinetics of Cefiderocol in Hospitalized Pediatric Participants (NCT04215991)	Single-dose phase: 3 months to less than 12 years with suspected or confirmed aerobic Gram-negative bacterial infections Multiple-dose phase: 3 months to less than 18 years with cUTI, HABP, or VABP	Single-dose phase: <34 kg 60 mg/kg; >34 kg 2 g over 3 h Multiple-dose phase: <34 kg 60 mg/kg; >34 kg 2 g q8h over 3 h	Recruiting	June 2024

TBD—to be decided; cUTI—complicated urinary tract infection; HABP—hospital-acquired bacterial pneumonia; VABP—ventilator-associated bacterial pneumonia. * Estimated.

**Table 3 microorganisms-11-01798-t003:** Clinical characteristics, management, and outcomes of ceftazidime/avibactam in infections due to MDR Gram-negative bacteria in the pediatric population.

Study	Study Design	Population	Infection	C/A MIC (μg/mL)	C/A Dosage and Duration	TDM (μg/mL)	AEs	Outcomes
**Bradley et al.** [98]	Single-blind, randomized, phase 2 study C/A compared with cefepime	Hospitalized children ≥3 months to <18 years with cUTI (C/A *n* = 67 cefepime *n* = 28)	cUTI, including acute pyelonephritis *E. coli* (90.7%)	NA	C/A doses q8h over 2 h: ≥3 months to <6 months: 40/10 mg/kg ≥6 months to <18 year (<40 kg): 50/12.5 mg/kg ≥6 years to <18 years (>40 kg): 2/0.5 g In the case of CrCl (≥30 to <50 mL/min): 50% dose reduction	Median values at 15 min 0.5–1.5 h 5–6 h C/A 78.35/13.20 47.10/6.88 6.91/0.88	Overall incidence: 53.7% (36/67) C/A 53.6% (15/28) cefepime	Favorable clinical response at TOC (mMITT): 88.9% (48/54) C/A 82.6% (19/23) cefepime
**Bradley et al.** [99]	Single-blind, randomized, phase 2 study C/A + MTZ compared with meropenem	≥3 months to <18 years with cIAI (C/A *n* = 61 meropenem *n* = 22)	cIAI: Most frequent origin: appendiceal perforation (85.2%) Most frequent pathogens: *E. coli* (84%) and *P. aeruginosa* (28%)	NA	C/A doses q8h over 2 h + MTZ q8h over 30 min ≥3 months to <6 months: 40/10 mg/kg ≥6 months to <18 years (<40 kg): 50/12.5 mg/kg ≥6 years to <18 years (>40 kg): 2/0.5 g In the case of CrCl (≥30 to <50 mL/min): 50% dose reduction	Median values at 15 min 0.5–1.5 h 5–6 h C/A 62.3/12.4 39.45/7.33 4.42/0.67	Overall incidence: 52.5% (32/61) C/A + MTZ 59.1% (13/22) meropenem	Favorable clinical response at TOC (ITT): 91.8% (56/61) C/A + MTZ 95.5% (21/22) meropenem
**Peruccio et al.** [66]	Retrospective, observational study	Children with hematological malignancies (*n* = 21) (subgroup patient characteristics NA)	MDR Gram-negative infections (subgroup microbiological characteristics NA)	NA	50/12.5 mg/kg q8h for a median of 14 days (range 6–19 days)	NA	Subgroup description of AEs NA	Subgroup outcome analysis NA
**Wang et al.** [97]	Retrospective, observational study	Children with liver transplant (*n* = 6)	Intraperitoneal infections (6/6) and BSI (5/6) CR-*K. pneumonia* (6/6) CR-*E. coli* (1/6)	*K. pneumonia*: 1 (4/6), 2 (2/6) *E.Coli*: NA	50/12.5 mg/kg q8h	NA	Vomiting (1/6), skin rash (1/6), increase in γ-GT (2/6) or ALP (3/6)	Clinical and microbiological resolution (6/6)
**Iosifidis et al.** [91]	Case series	Patients aged from 13 days to 4.5 years (*n* = 8)	XDR or PDR *K. pneumoniae* infections Possible or proven BSI (7/8)	Etest: 0.75 (*n* = 5) Disk: 17 mm (*n* = 1), 20 mm (*n* = 1), 22 mm (*n* = 2)	50/12.5 mg/kg q8h (*n* = 7) 25/6.25 mg/kg q8h due to AKI and CVVH (*n* = 1)	NA	No serious AEs, discontinuation or dose modification due to any AEs.	Clinical microbiologic and response (*n* = 8/8)
**Alamarat et al.** [95]	Case report	15-year-old male with chronic osteomyelitis	XDR-*P. aeruginosa* and ESBL-producing *K. pneumoniae*	*P. aeruginosa* >256 *K. pneumonia* 0.38	2/0.5 g q8h + ATM 2 g q8h	NA	None	Clinical and microbiological failure and switch to cefiderocol
**Hobson et al.** [100]	Case report	3-year-old female with relapse of LAL	NDM-1-producing *Morganella morganii*	C/A > 256 ATM: 4 C/A + ATM 0.016	120/30 mg/kg/day + ATM 100 mg/kg/day	NA	NA	Clinical and microbiological resolution
**Cowart et al.** [101]	Case report	11-year-old female with cystic fibrosis	*S. maltophilia* pulmonary exacerbation	C/A >256 μg/mL ATM >256 μg/mL C/A + ATM 8 μg/mL	150/37.5 mg/kg/day over 2 h + ATM 200 mg/kg/day over 3 h After TDM adjustments: 200/50 mg/kg/day in CI + ATM 200 mg/kg/day	1 h after bolus: C 6 ATM 67.6 CI: C 50.2 ATM 96.5	None	Improvement of symptoms and lung examination observed but a progressive decline in lung function persisted
**Ren et al.** [92]	Case report	4-year-old female with intracranial SOL	ESBL-*E. coli* post-neurosurgical meningitis and abscess	<0.125	1/0.25 g q8h for 45 days + meropenem	CSF post-infusion 3 h: 15.6/4 5 h: 7.1/2.1 7 h: 3.5/2.1 Serum post-infusion: 3 h 57/11.3 5 h 25.8/4.5	None	Clinical and microbiological resolution
**Tamma et al.** [94]	Case report	2-month female infant with congenital diaphragmatic hernia surgically repaired	*Burkholderia cepacia* complex BSI	2	120/30 mg/kg/day in CI for 6 weeks	NA	None	Clinical and microbiological resolution
**Coskun et al.** [96]	Case report	25-day-old preterm neonate (27 gestational weeks)	XDR-*K. pneumoniae* urinary tract infection	NA	50/12.5 mg/kg q8h for 10 days	NA	Glycosuria	Clinical and microbiological resolution
**Almangour et al.** [93]	Case report	2-year-old male delivered at 26 weeks with hydrocephalus	MDR-*P. aeruginosa* infection of the ventriculoperitoneal shunt	2	50/12.5 mg/kg/dose q8h for 21 days + IVT colistin for 14 days	NA	None	Clinical and microbiological resolution

BSI—bloodstream infection; C/A—ceftazidime/avibactam; C—ceftazidime; A—avibactam; MTZ— metronidazole; AEs—adverse effects; SAEs—serious adverse events; TDM—therapeutic drug monitoring; CF—cystic fibrosis; IAI—intra-abdominal infection; MIC—minimum inhibitor concentration; NA—not available; C_max_—peak concentration; C_min_—trough concentration; micro-ITT—microbiologic intent to treat; ITT—intent to treat; TOC—test of cure; MTZ—metronidazole; XDR—extensively drug resistant; ATM—aztreonam; CI—continuous infusion; SOL—space occupying lesion; LAL—lymphoblastic acute leukemia; γ-GT—gamma-glutamyltransferase; ALP—alkaline phosphatase; CR—carbapenem resistant; IVT—intraventricular.

## Data Availability

Not applicable.

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
