# Peer review of "Novel Beta Lactam Antibiotics for the Treatment of Multidrug-Resistant Gram-Negative Infections in Children: A Narrative Review"

_microorganisms, 2023, doi:10.3390/microorganisms11071798_

Round 1

Reviewer 1 Report

Dear author the narrative review discusses the use of novel beta-lactam antibiotics for the treatment of multidrug-resistant Gram-negative infections in children. The review emphasizes the challenges faced by clinicians in selecting appropriate antimicrobial therapy due to the proliferation of drug-resistant bacteria. It highlights the urgent need for new therapeutic options.

Table 1 is hard to read!

Please use abreviation correct!

However, a recent study specifically analyzing isolates from pediatric patients with cystic fibrosis (CF) found that up to 53% of the isolates were resistant to ceftazidime/avibactam [45]. This indicates that the efficacy of ceftazidime/avibactam may be lower in the context of CF-associated infections in the pediatric population. Please add more data!

In conclusion By conducting pediatric PK/PD studies, researchers and clinicians can optimize the use of novel antimicrobials, ensuring their effectiveness and safety in treating MDR infections in children. It is an ongoing area of research and collaboration between scientists, regulatory bodies, and healthcare professionals to address this urgent need and improve the management of MDR infections in pediatric patients. please add 3 short conclusion usefull fr clinicians!

I noticed a few minor grammatical errors, such as incorrect verb tense agreement and punctuation inconsistencies. A thorough proofreading session would help eliminate these errors and improve the overall polish of the work.

Can be accepted after correction!

Author Response

> Dear author the narrative review discusses the use of novel beta-lactam antibiotics for the treatment of multidrug-resistant Gram-negative infections in children. The review emphasizes the challenges faced by clinicians in selecting appropriate antimicrobial therapy due to the proliferation of drug-resistant bacteria. It highlights the urgent need for new therapeutic options.
Table 1 is hard to read!
Please use abbreviation correct!
As suggested by reviewer we modified the Table 1 in order to make it more readable and provided to correct the abbreviations.

> However, a recent study specifically analyzing isolates from pediatric patients with cystic fibrosis (CF) found that up to 53% of the isolates were resistant to ceftazidime/avibactam [45]. This indicates that the efficacy of ceftazidime/avibactam may be lower in the context of CF-associated infections in the pediatric population. Please add more data!

We thank the reviewer for the comment that allows us to improve the text.
We added in the revised manuscript more details and data about the abovementioned study (line 245) and we specified that this finding might be influenced by local epidemiology. We did not find additional data about paediatric patients with CF and ceftazidime/avibactam treatment.

> In conclusion By conducting pediatric PK/PD studies, researchers and clinicians can optimize the use of novel antimicrobials, ensuring their effectiveness and safety in treating MDR infections in children. It is an ongoing area of research and collaboration between scientists, regulatory bodies, and healthcare professionals to address this urgent need and improve the management of MDR infections in pediatric patients. please add 3 short conclusion usefull fr clinicians!

In consideration of the reviewer’s comment we introduced in the new version of manuscript more conclusions/suggestions for the clinicians, also considering the reviewer’s comment. In particular we underlined the importance to work as a team of different medical professions, to take in the account ‘patient’s variability’ in term of clinical manifestations and genetic background, o consider the local epidemiology and at the same time the experience that the Center has in the field. We think that these aspects are important because correlated to the efficacy of treatment.

> Comments on the Quality of English Language
I noticed a few minor grammatical errors, such as incorrect verb tense agreement and punctuation inconsistencies. A thorough proofreading session would help eliminate these errors and improve the overall polish of the work.
Can be accepted after correction!
Thank you the reviewer for the positive comments. We reviewed the manuscript carefully.

Reviewer 2 Report

Here the authors performed a literature review to compile the data available for the use of new antibiotics or antibiotic combinations against multidrug resistant bacteria primarily carbapenem-resistant Enterobacterales amongst pediatric population. The rise of antibiotic resistance, scarcity in clinical and pharmacokinetic studies and the lack of new treatments for the newborn and infant populations are increasingly being recognized in the scientific and clinical world. Several studies have been published in recent years to address these issues and the current review is a good addition. It is well written, except for a few minor comments that should be addressed.

1.     Line 35-36: Based on the reference cited, 37 -60% antibiotic exposure amongst hospitalized pediatric patients are global estimates. It is suggested that the authors specify this information to highlight the significance and the worldwide nature of the problem.

2.     Line 195: The data available on ceftolozane/tazobactam efficacy and usage are indeed promising. However as highlighted by the authors, based on the pediatric patient sample size of the studies, it is still premature to state that they are "proved to be safe and effective". The authors should re-word this sentence.

3. There are some typos. The authors should therefore do a thorough review of the spelling. Abbreviations like C/T have been mentioned in fig legends however, they should also be specified at their first mention in the text.

There are some typos. The authors should therefore do a thorough review of the spelling. 

Author Response

Here the authors performed a literature review to compile the data available for the use of new antibiotics or antibiotic combinations against multidrug resistant bacteria primarily carbapenem-resistant Enterobacterales amongst pediatric population. The rise of antibiotic resistance, scarcity in clinical and pharmacokinetic studies and the lack of new treatments for the newborn and infant populations are increasingly being recognized in the scientific and clinical world. Several studies have been published in recent years to address these issues and the current review is a good addition. It is well written, except for a few minor comments that should be addressed.

  1. Line 35-36: Based on the reference cited, 37 -60% antibiotic exposure amongst hospitalized pediatric patients are global estimates. It is suggested that the authors specify this information to highlight the significance and the worldwide nature of the problem.
As suggested by reviewer we specified in the text the worldwide relevance that has the use of the antibiotics.

  1. Line 195: The data available on ceftolozane/tazobactam efficacy and usage are indeed promising. However as highlighted by the authors, based on the pediatric patient sample size of the studies, it is still premature to state that they are "proved to be safe and effective". The authors should re-word this sentence.
We thank the reviewer for the suggestion and re- wrote the sentence.
  1. There are some typos. The authors should therefore do a thorough review of the spelling. Abbreviations like C/T have been mentioned in fig legends however, they should also be specified at their first mention in the text.
As suggested by reviewer we mentioned in the introduction the C/T abbreviation.

Comments on the Quality of English Language
There are some typos. The authors should therefore do a thorough review of the spelling.

As suggested by reviewer we reviewed the English, the grammar and the spelling of the text.

Reviewer 3 Report

The article presented to me for review, "Novel Beta Lactam Antibiotics for the Treatment of Multidrug Resistant Gram-negative Infections in Children: a Narrative Review" concerns the important medical issue of infections with bacteria resistant to routine antibiotics. It is important to emphasize the importance of the topic which the authors have addressed. Antibiotic resistance is one of the greatest challenges of current medicine. The topic discussed is obviously very important for every pediatrician. 

In 5 chapters, the authors discussed the results of the most important studies on the antibiotics under discuss.

In their paper, the authors included 3 tables

In conclusion, they drew the right conclusions from the analyzed available material. 

In the article, the authors cited 157 items of current medical literature. 

In my opinion, the authors have made an in-depth analysis of the topic being discussed. They have done a gigantic job. They have presented an overview of the results of the most important studies on the topic under discussion in a clear and accessible manner. 

In my opinion, the work can be accepted for publication in its present form

Author Response

The article presented to me for review, "Novel Beta Lactam Antibiotics for the Treatment of Multidrug Resistant Gram-negative Infections in Children: a Narrative Review" concerns the important medical issue of infections with bacteria resistant to routine antibiotics. It is important to emphasize the importance of the topic which the authors have addressed. Antibiotic resistance is one of the greatest challenges of current medicine. The topic discussed is obviously very important for every pediatrician.

In 5 chapters, the authors discussed the results of the most important studies on the antibiotics under discuss.

In their paper, the authors included 3 tables

In conclusion, they drew the right conclusions from the analyzed available material.

In the article, the authors cited 157 items of current medical literature.

In my opinion, the authors have made an in-depth analysis of the topic being discussed. They have done a gigantic job. They have presented an overview of the results of the most important studies on the topic under discussion in a clear and accessible manner.

In my opinion, the work can be accepted for publication in its present form

We thank the reviewer for the positive comment about our work.